# Effects of Single-arc Blade Profile Length on the Performance of a Forward Multiblade Fan

**Yikun Wei [1], Cunlie Ying [2], Jun Xu [2], Wenbin Cao [2], Zhengdao Wang [1,\*,†] and Zuchao Zhu [1,\*,†]**

[1] National-Provincial Joint Engineering Laboratory for Fluid Transmission System Technology, Faculty of Mechanical Engineering and Automation, Zhejiang Sci-Tech University, Hangzhou 310018, China; yikunwei@zstu.edu.cn

[2] Zhejiang Yilida Ventilator Co., Ltd., Taizhou 318056, China; yingcunlie@yilida.com (C.Y.); xujun@yilida.com (J.X.); caowenbin@yilida.com (W.C.)

\* Correspondence: dao@zstu.edu.cn (Z.W.); zhuzuchao@zstu.edu.cn (Z.Z.); Tel.: +86-13906507577 (Z.Z.)

† These authors contributed equally to this work.

**Abstract:** The effects of single-arc blade profile length on the performance of a forward multiblade fan are investigated in this paper by computational fluid dynamics and experimental measurement. The present work emphasizes that the use of a properly reduced blade inlet angle ($\beta_{1A}$) and properly improved blade outlet angle ($\beta_{2A}$) is to increase the length blade profile, which suggests a good physical understanding of internal complex flow characteristics and the aerodynamic performance of the fan. Numerical results indicate that the gradient of the absolute velocity among the blades in model-L (reducing the blade inlet angle and improving blade outlet angle) is clearly lower than that of the baseline model and model-S (improving the blade inlet angle and reducing blade outlet angle), where a number of secondary flows arise on the exit surface of baseline model and model-S. However, no secondary flow occurs in model-L, and the flow loss at the exit surface of the volute (scroll-shaped flow patterns) for model-L is obviously lower than that of the baseline model at the design point. The comparison of the test results further shows that to improve the blade profile length is to increase the static pressure and the efficiency of the static pressure, since the improved static pressure of the model-L rises as much as 22.5 Pa and 26.2%, and the improved static pressure efficiency of the model-L rises as much as 5 % at the design flow rates. It is further indicated that increasing the blade working area provides significant physical insight into increasing the static pressure, total pressure, the efficiency of the static pressure and the total pressure efficiency.

**Keywords:** performance; blade profile length; forward multiblade fan; numerical simulation; experimental test

## 1. Introduction

It is well known that centrifugal fans have been widely adopted to industrial application, such as cooling units in air-conditioning, heating, ventilating and air conditioning systems, and home appliance machines [1–5]. In general, it is essential for the centrifugal fan to improve its aerodynamic performance to achieve the purpose of energy conservation [5–9]. Many centrifugal fans with moderate efficiency and low aero-acoustic performances still need to be improved by making use of today's technology. Velarde and Ballesteros [10] reported that to improve the blade number, tongue length and scroll contour is to improve the aerodynamic performance of a centrifugal blower. Younsi et al. [11] investigated that the internal flow of the blower for various rotational frequencies effects the performance characteristics of a turbo blower in a refuse collecting system. Lin and Huang [12] investigated that the performance of a centrifugal fan is affected by changing the blade inlet angle and the housing outlet geometry through experimental and computational methods.

Lun et al. [13] evaluated the total pressure and efficiency of a centrifugal fan by changing the impeller gap and blade inlet angle. Although varying the blade's outlet angle does not appear to be very advantageous herein, it certainly shows the potential benefits of such a modification, especially in cases where the fan design is not optimal and demands vigorous enhancements [14]. The transient surface of the models with different outlet angles can be found [15–17]. Computational fluid dynamics (CFD) results were implemented to provide insight into revealing flow loss in the internal flow of fan. Thus, the prediction method of CFD is represented to study the internal flow mechanisms in the fan. The section of computational methodology includes solver descriptions of the Navier–Stokes and the used turbulence models associated, computational procedures for non-rotating and rotating components of the fan, gridding strategy and a study of grid density, and flow parameters for the performance of predicting the fan [18,19]. To validate the predictions of CFD for the performance of fan data, comparisons with one-fifth-scale model fan test data are provided [20]. The trend is quite straightforward: Increasing the outlet angle leads to a higher pressure rise and torque across the whole operating range, and vice versa.

Complex turbulent structures due to rotating Coriolis force and narrow blade passage in the internal flow of a forward multiblade fan play a dominant role in directly affecting the performance of the fan. Once these complex turbulent internal flows of the fan have been described, and the underlying internal flow mechanisms have been understood, new open questions will emerge in the internal flow of the fan. How do we restrain turbulent flow to reduce flow loss in the internal flow of our fan? In the low to mid-range flow rates, changing the outlet angle does not considerably affect the static efficiency of the fan, whereas the ensuing effects in the higher range are significant, and the impellers with larger outlet angles show a better performance.

Based on the above discussions, our study mainly focuses on the effects of the outlet angle of the impeller blades on the aerodynamic performance of the centrifugal fan for potential energy conservation. The origins and effects of a single-arc blade profile length on the performance of a forward multiblade fan are discussed with the emphasis on the understandings of the physical mechanisms of internal complex flow characteristics. Our work indicates that by reducing the blade inlet angle ($\beta_{1A}$, $\beta_{1A} > 60°$), improving the blade outlet angle ($\beta_{2A}$, $\beta_{2A} < 175°$) and increasing the blade's working area may provide significant physical insight into increasing the static pressure and the efficiency of this static pressure. It gives insight into improving the aerodynamic load of fans to achieve the purpose of energy conservation. The performance of numerical results is quantitatively and qualitatively compared with that of the experimental results. This paper is organized as follows. In Section 2, governing equations and the numerical method will be briefly described at first. After that, the detailed numerical and experimental results are discussed. In final, some conclusions are addressed.

## 2. Governing Equations and Numerical Method

In the following section, the three-dimensional equation of fluid dynamics is implemented in fluid flow analysis of the computational domain [15,16]. The incompressible Reynolds Averaged Navier–Stokes (RANS) Turbulence Models are discretized by the finite volume method [16]. The equation of fluid dynamics is:

$$\frac{\partial(\rho u_i)}{\partial x_i} = 0 \tag{1}$$

$$\frac{\partial(\rho u_j u_i)}{\partial x_j} = f_i - \frac{\partial P^*}{\partial x_i} + \frac{\partial\left[\mu_e\left(\frac{\partial u_i}{\partial x_j} + \frac{\partial u_j}{\partial x_i}\right)\right]}{\partial x_j} \tag{2}$$

in which $\rho$ is fluid density, $x_i$ and $x_j$ denote the components $x$, $y$ and $z$ in the rectangular coordinate system in three directions, $u_i$ and $u_j$ are the average relative velocity components $u$, $v$ and $w$, $P^*$ denotes the reduced pressure, $f_i$ is the volume force component, and $\mu_e$ is the efficient viscosity coefficient [17,21].

The Re-Normalization Group (RNG) k-epsilon model is implemented in steady flow. The empirical formula of turbulence model RNG k-epsilon is as follows [22,23].

$$\frac{\partial}{\partial t}(\rho k) + \frac{\partial}{\partial x_i}(\rho k u_i)_k = \frac{\partial}{\partial x_j}\left(\alpha_k \mu_{eff}\frac{\partial k}{\partial x_j}\right) + G_k + G_b - \rho\varepsilon - Y_M + S_k \tag{3}$$

$$\frac{\partial}{\partial t}(\rho\varepsilon) + \frac{\partial}{\partial x_i}(\rho\varepsilon u_i) = \frac{\partial}{\partial x_j}\left(\alpha_\varepsilon \mu_{eff}\frac{\partial\varepsilon}{\partial x_j}\right) + C_{1\varepsilon}\frac{\varepsilon}{k}(G_k + G_{3\varepsilon}G_b) - C_{2\varepsilon}\rho\frac{\varepsilon^2}{k} - R_\varepsilon + S_\varepsilon \tag{4}$$

in which $G_k$ is the turbulent kinetic energy of laminar velocity gradient, $G_b$ denotes the turbulent kinetic energy, $Y_M$ is the effect of compressible turbulent fluctuation expansion, $C_{1\varepsilon}$, $C_{2\varepsilon}$ and $G_{3\varepsilon}$ are constants, $\alpha_k$ and $\alpha_\varepsilon$ are the turbulent Prandtl numbers of the $k$ equation and $\varepsilon$ equation, respectively. The above equations are solved by implementing the finite volume method. In first, the time discretization of the Navier–Stokes equations coupled with some governing equation for a pollutant is implemented in this paper. The spatial schemes are used to account for convective terms, viscous terms, mean pressure gradient effects and source terms associated with momentum or scalar equations, focusing on triangular meshes [20,24].

## 3. Computational Model and Grid of Fan

The forward multiblade centrifugal fan with a double-width double-inlet impeller is implemented in the current investigation. Figure 1 shows the double inlet radial fan with a forward curved blade. It consists of the impeller hub plate, volute, the mid-disc of impeller and the symmetry plane. The blade shape of test cases and some key parameters of the baseline model are presented in Figure 2. The complex vortex structure of the baseline model on the internal flow and performance is referred to in the previous work by Lun [13]. As shown in Figure 2, we can obtain that the blade profile of the baseline model is a single circular arc with the length of 10.904 mm; model-L refers to the model with a longer blade, and model-S with a shorter blade. Plotted in Figure 2, it is seen that the blade profile of model-L (reducing the blade inlet angle and improving blade outlet angle) is 150% that of the baseline model, and model-S (improving the blade inlet angle and reducing blade outlet angle) is 80%. The blade profiles of both modified models are still single circular arcs which are shown in Figure 2. The blade length of model-S, the baseline model and model-L increase successively in Figure 2.

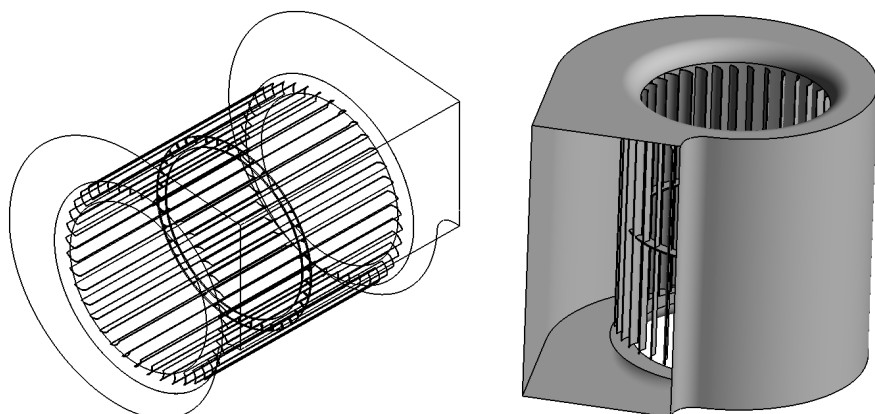

**Figure 1.** Example of double inlet radial fan with forward curved blade.

Table 1 shows the blade inlet angle and blade outlet angle of three models. As shown in Table 1, one can see that the blade inlet angles of the baseline model, model-S and model-L, are 84.83°, 91.27° and 71.34°, respectively, and the blade outlet angles of the baseline model, model-S and model-L are 152.36°, 146.69° and 170.13°, respectively. The numerical model of the radial fan is divided into five independent domains, as shown in Figure 3. Plotted in Figure 3, the impeller domain (domain 3) is set

as the rotor and others are set as stators, while each domain is connected by an interface. The structured mesh is implemented for the whole fan model, and details of the numerical meshes of the volute and static pressure at a different number of grids are given in Figure 4. We can obtain that the change rate of static pressure is less than 1% at different mesh numbers. Thus, we can ignore the static pressure change. The numerical results are not affected by the number of grids. In all simulations, the ANSYS CFX software (CFX 16.0, Ansys, Canonsburg, PA 15317, USA) is implemented to study the internal complex flow of the forward multiblade centrifugal fan. The Multi Frames of Reference (MFR) technique is one such interface model. The MFR approach obtained by ANSYS CFX 16.0 is implemented to calculate the part of the impeller in the reference frame and the other parts in the stationary reference frame [12,13]. An interface is used at each junction where the frame change of reference takes place. There are two kinds of interface techniques available to exchange the information between the different frames of reference [12,13].

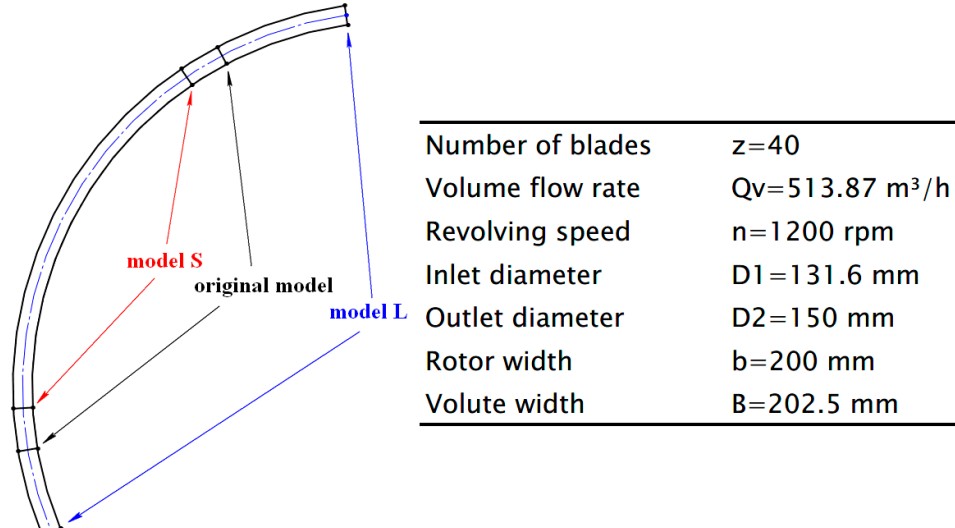

| | |
|---|---|
| Number of blades | z=40 |
| Volume flow rate | Qv=513.87 m³/h |
| Revolving speed | n=1200 rpm |
| Inlet diameter | D1=131.6 mm |
| Outlet diameter | D2=150 mm |
| Rotor width | b=200 mm |
| Volute width | B=202.5 mm |

**Figure 2.** Blade shape of test cases and geometrical parameters of the baseline model.

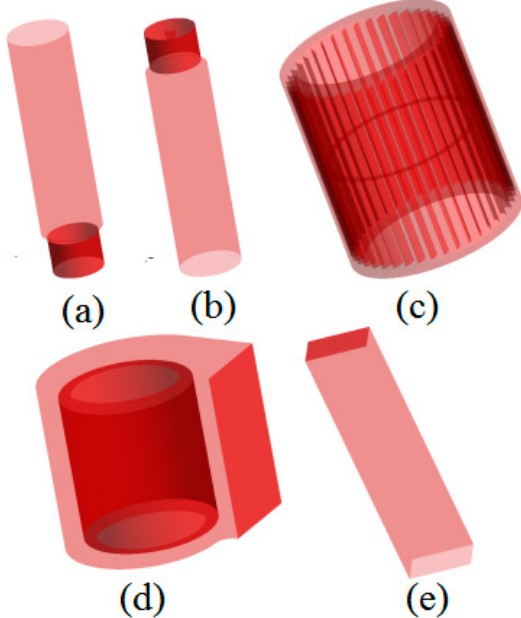

**Figure 3.** Domain composition: Inlet left (**a**), inlet right (**b**), impeller (**c**), volute (**d**) and outlet (**e**). The domain interface is colored in deep red.

**Table 1.** Blade inlet angle and blade outlet angle of the three models.

|  | Baseline Model | Model-S | Model-L |
|---|---|---|---|
| Blade inlet angle ($\beta_{2A}$) | 84.83° | 91.27° | 71.34° |
| Blade outlet angle ($\beta_{1A}$) | 152.36° | 146.69° | 170.13° |

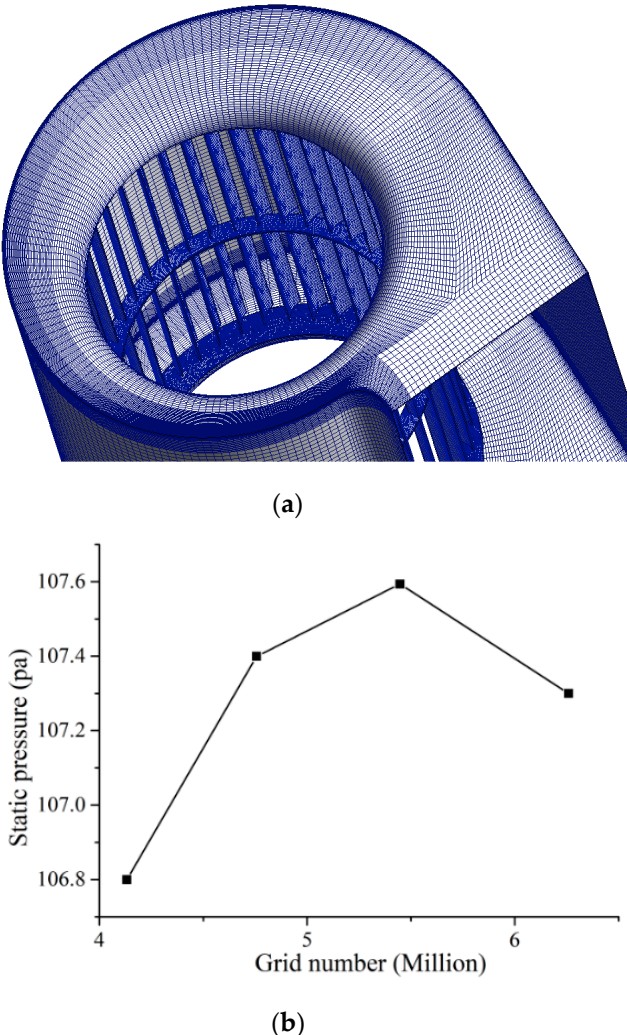

(**a**)

(**b**)

**Figure 4.** Detail of the numerical mesh and Static pressure at different number of grids. (**a**) numerical meshes of the volute; (**b**) static pressure at different number of grids.

## 4. Results and Discussions

Finite element methods [24–27] and the finite volume method [28,29] are very effective tools to solve some partial differential equations (PDEs) on complex geometries, which is applied in a wide range of engineering and biomedical disciplines [30–34]. In this paper, the finite volume method is implemented to calculate the NS equation. Numerical simulations are carried out using steady RANS based on the RNG k-ε turbulence model in our studies. The empirical formula of the turbulence model RNG k-epsilon is introduced in detail in Section 2. The mass flow rate is implemented as the inlet condition, and static pressure is implemented as the outlet condition. The wall condition is no slip boundary. It is well known that a moving fluid in contact with a solid body will not have any velocity relative to the body at the contact surface. This condition of not slipping over a solid surface has to be satisfied by a moving fluid. The second order upwind discretized form is used for all cases in incompressible flows [21,22]. The second order upwind discretized form is mainly based upon a

coupled solution of the nonlinear finite difference equations according to the multigrid technique. Calculations have been made of the driven turbulent flow for several Reynolds numbers and finite difference grids. In comparison with the hybrid discretized form, the second order upwind discretized form is somewhat more accurate, but it is not monotonically accurate with mesh refinement.

## 4.1. Numerical Verification

In order to confirm the accuracy of numerical results, the baseline model is implemented to study its static pressure and the efficiency of this static pressure at various flow rates. Figure 5 shows the comparison of static pressure as well as the static pressure efficiency between numerical results and experimental results. As shown in Figure 5, one can clearly see that the static pressure (P, Pa) decreases with the increase of flow rate (Q, m$^3$/h) in both the experiment and numerical simulation. It is also obtained that the static pressure curve and the static pressure efficiency curve of the baseline model is well consistent with the experimental results at various flow rates, which demonstrates that the numerical accuracy is reliable.

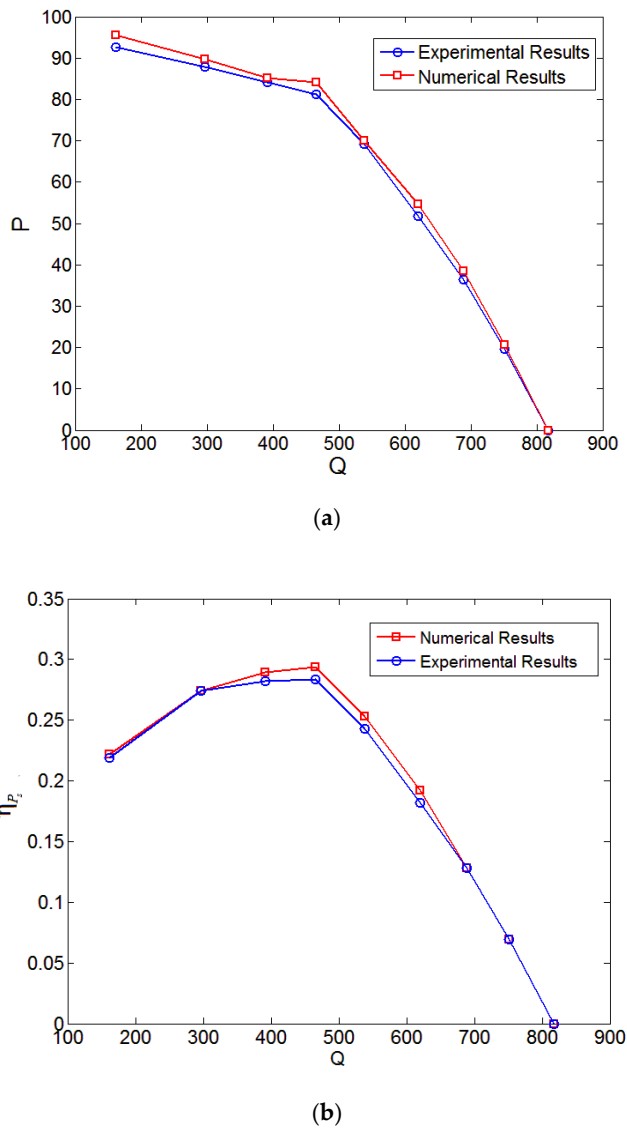

**Figure 5.** Comparison of static pressure and the static pressure efficiency between numerical results and experimental results. (**a**) curve of static pressure -flow (**b**) curve of static pressure efficiency-flow.

### 4.2. Complex Internal Flow in Different Forward Multiblade Fans

In the following section, complex internal flow in different forward multiblade fans will be introduced. The analysis refers to the centrifugal fan operating at three mass flow rate conditions of 0.0403 kg/s ($0.235Q_n$), 0.1713 kg/s ($Q_n$) and 0.2809 kg/s ($1.640Q_n$). Figure 6 shows the overview of the flow behavior within the impeller with the absolute velocity contours. Plotted in Figure 6, $S_1$, $S_2$ and $S_3$ represent the cutting planes at the hub, at the mid-span and at the tip of the impeller, respectively.

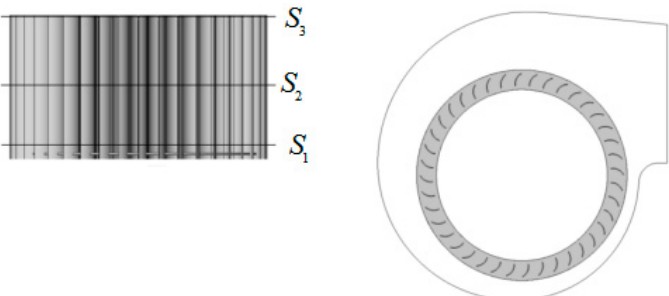

**Figure 6.** Different heights within the impeller.

Figure 7 shows the absolute velocity at different heights within the impeller. As shown in Figure 7, it is obviously seen that the gradient of the absolute velocity among the blades decreases with the successive increase of the blade length of model-S, the baseline model and model-L at the cross section of $S_1$, $S_2$ and $S_3$, which mainly indicates that the size of the flow separation among the blades becomes smaller with increasing the blade outlet angle. We can see that on the hub surface ($S_1$), the velocity difference between inlet and outlet of the impeller increases, and the velocity gradient in the impeller passage of model-L is more uniform than that of others. The area of the low velocity region at the inlet of the impeller decreases because of lengthening the blade in model-L. The average velocity decreases with the increase of axial height in the impeller passage. The main reason is that the flow in the passage of the tip cutting plane ($S_3$) is greatly affected by the inflow near the inlet of the volute.

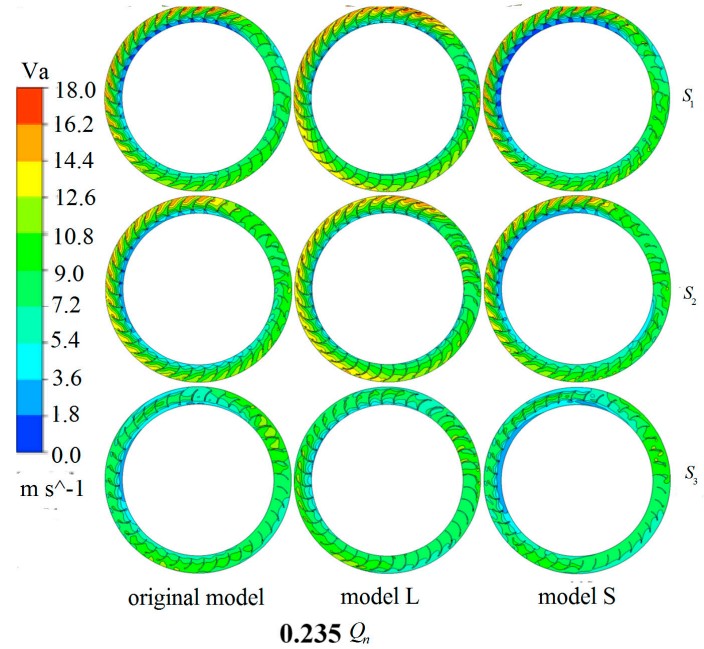

**Figure 7.** Absolute velocity at different heights within the impeller at $0.235Q_n$.

Figure 8 shows the absolute velocity at different heights within the impeller at design point $Q_n$. It is clearly observed that at the design point, the high velocity region exists at the impeller outlet of model-L at the hub plane ($S_1$) and midspan ($S_2$). However, model-L possesses more uniform velocity distribution in the impeller passage than that of the baseline model and model-S. Figure 9 shows the absolute velocity at different heights within the impeller at $1.64Q_n$. One can see that the high absolute velocity region in model-L is obviously larger than that of the baseline model and model-S. A large number of low absolute velocity regions occur at the inlet of blades in the baseline model and model-S at sections of $S_1$ and $S_2$. Nevertheless, the gradient of the absolute velocity among the blades in model-L is clearly lower than that of the baseline model and model-S.

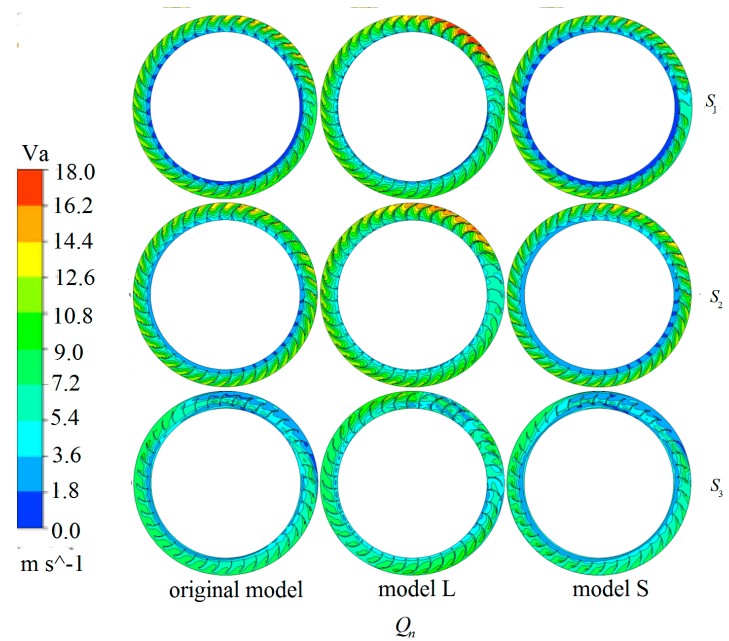

**Figure 8.** Absolute velocity at different heights within the impeller at $Q_n$.

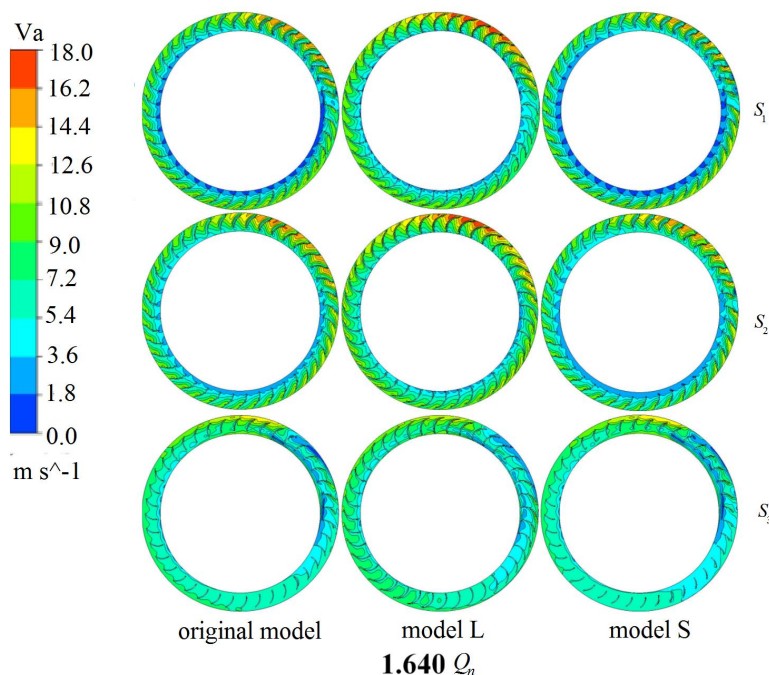

**Figure 9.** Absolute velocity at different heights within the impeller at $1.640Q_n$.

From Figure 7 to Figure 9, it is also obtained that the high velocity region at the impeller outlet moves toward the outlet of volute, and the difference of velocity in the blade passage increases accordingly with the increase of the mass flow rate. It is also noted that the high absolute velocity region in model-L is obviously higher than that of the baseline model and model-S. However, the gradient of the absolute velocity among the blades in model-L is clearly lower than that of the baseline model and model-S. It is further found that the gradient of the absolute velocity among the blades decreases in model-L, which brings about the decreasing size of the flow separation among the blades.

Figure 10 delineates the turbulence kinetic energy (TKE) of impeller outlet at $0.235Q_n$. It is clearly observed that the high turbulence kinetic energy area in the baseline model occurs at the inlet of the volute. Meanwhile, the high turbulence kinetic energy area in model-L and model-S also occurs at the inlet of the volute. It is noted that for the turbulence kinetic energy, no significant difference occurs near the inlet of the volute in the baseline model, model-L and model-S.

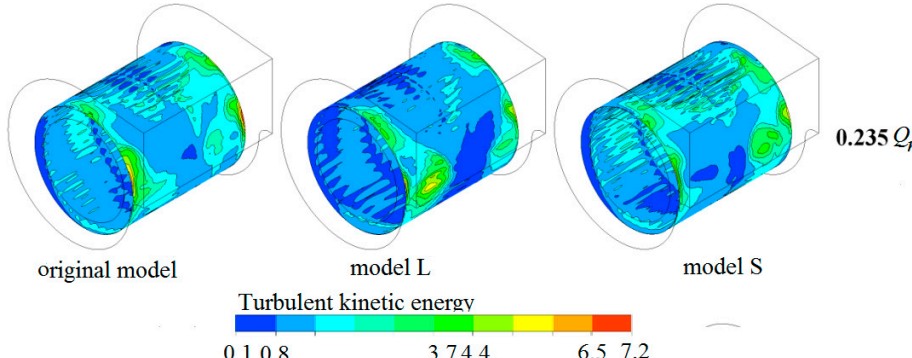

**Figure 10.** Distribution of turbulence kinetic energy of the impeller at $0.235Q_n$.

Figure 11 shows the TKE of the impeller outlet at the design point $Q_n$. One can obviously see that at design point $Q_n$, the turbulence kinetic energy area near the inlet of the volute for model-L is lower than that of baseline model and model-S. It is also noted that the TKE at the impeller outlet of model-L and model-S is obviously reduced as compared with that of the baseline model and model-S.

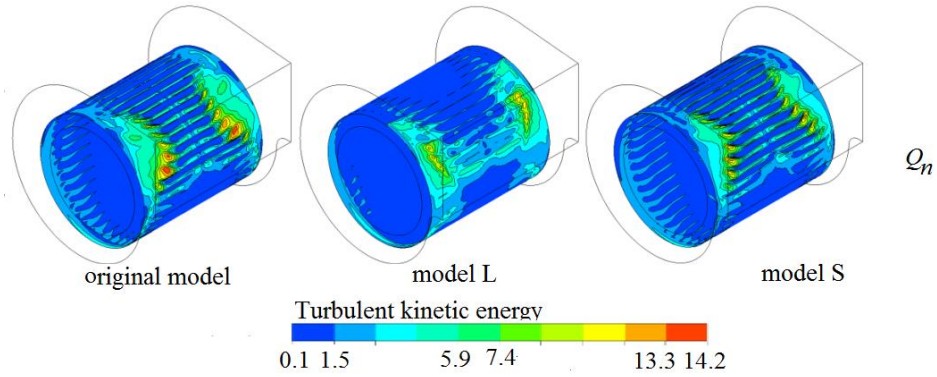

**Figure 11.** Distribution of turbulence kinetic energy of the impeller at $Q_n$.

Figure 12 demonstrates the TKE of the impeller outlet at $1.64Q_n$. The variation tendency at $1.64Q_n$ for the turbulence kinetic energy area near the inlet of the volute is same as that of the design point $Q_n$. From Figure 10 to Figure 12, it is obtained that model-L has a better status as a whole, in not only reducing the peak of TKE at the impeller outlet, but also making the TKE change more uniformly along the circumferential direction of impeller outlet, especially in the area far from the volute outlet.

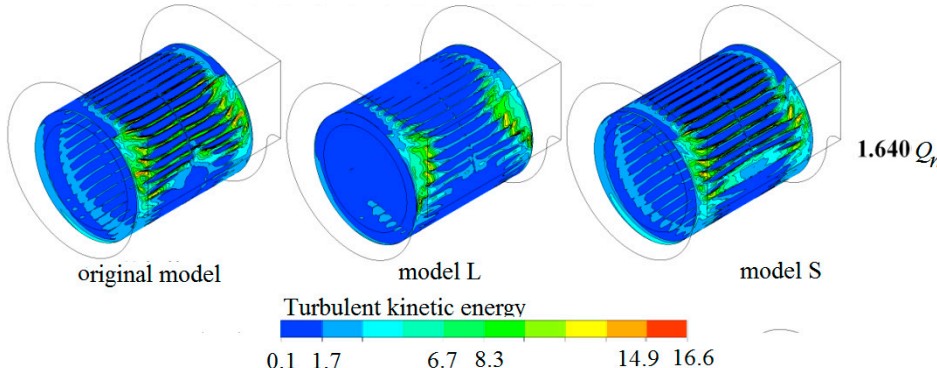

**Figure 12.** Distribution of turbulence kinetic energy of the impeller at 1.640$Q_n$.

Figure 13 shows the static pressure distribution on the surface of the volute tongue at 0.235$Q_n$. One can see that a low-pressure region and a high-pressure region appear on the surface of the volute tongue along the flow direction in each model at the condition of the lower flow rate, and the distribution range is large along the axial direction. For the model-L, the high-pressure region mainly locates at the middle part of tongue surface; the pressure value near middle part of tongue surface is obviously higher than that of the model-S and baseline model at 0.235$Q_n$.

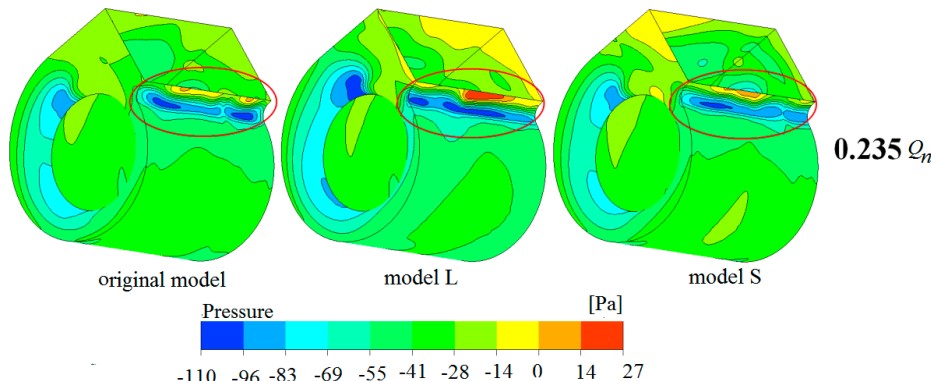

**Figure 13.** Static pressure distribution on the surface of the volute tongue at 0.235$Q_n$.

Figures 14 and 15 show the static pressure distribution on the surface of the volute tongue at design points $Q_n$ and 1.64$Q_n$, respectively. Plotted in Figure 14, it is observed that the low-pressure region disappears at design point $Q_n$, and is replaced by a high-pressure region mainly in the middle of the volute tongue, which occupies about one third of the area. As shown in Figure 15, it is noted that the area of the high-pressure region on the middle of volute tongue surface increases at a high flow rate condition while the low-pressure region appears on both sides of the tongue.

Figure 16 shows the velocity profile (along the circumferential direction) near the blade trailing edge of impeller exit for the baseline model, model-S and model-L at different flow rates. As shown in this figure, it is observed that at 0.235$Q_n$, the velocity distribution at the outlet of each model impeller varies slightly and seemed uniform along the circumferential direction. Little difference appears for the peak velocity of each model; the velocity slowly decreases near the volute outlet and the volute tongue area. It is also obtained that at design rate ($Q_n$) and high flow rate (1.640$Q_n$), the velocity of all models greatly increases near the outlet of the volute and then rapidly decreases near the volute tongue, and the difference between the highest and the lowest velocity magnitude on the profile increases along the circumferential direction. The velocity of model-L is the highest near the volute outlet, but the amplitude of the velocity fluctuation is the smallest over the whole circumferential distance. In other words, the velocity profile (along the circumferential direction) near the blade trailing edge

of the impeller exit obviously increases with the increase of the blade length at various flow rates. Nevertheless, the velocity amplitude of fluctuation decreases with the increase of the blade length.

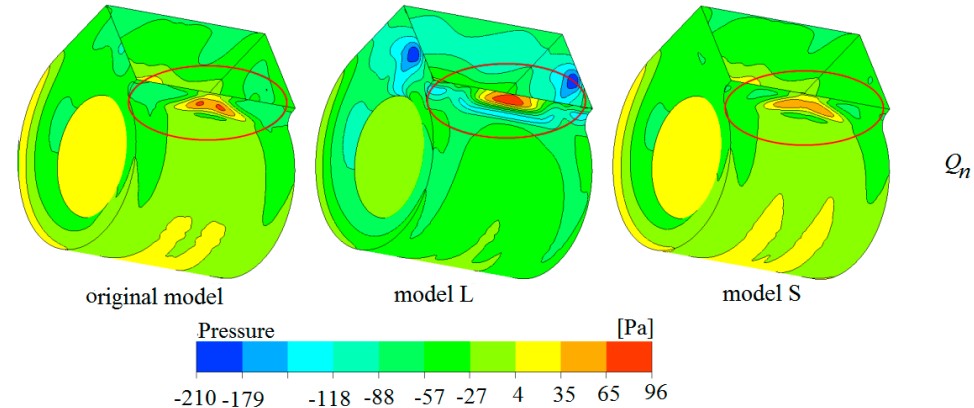

**Figure 14.** Static pressure distribution on the surface of the volute tongue at $Q_n$.

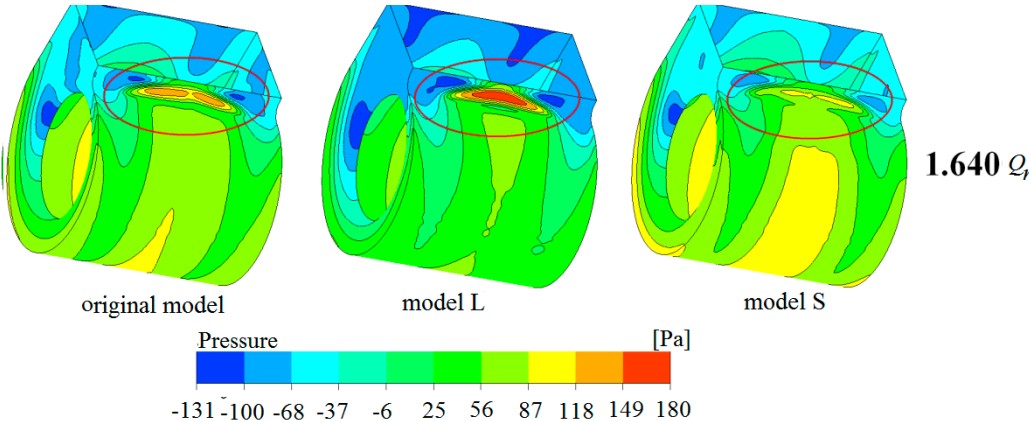

**Figure 15.** Static pressure distribution on the surface of the volute tongue at $1.640Q_n$.

The streamline at the exit surface of the volute is shown in Figure 17. One can see that at a lower flow rate condition, a number of small vortices form on both sides of the volute exit in the baseline model, where one small vortex forms on the bottom of model-S and no small vortex forms in model-L, which indicates that the flow loss at the exit surface of volute for model-L is obviously lower than that of the baseline model. At design point, a number of secondary flows arise at the exit surface of the baseline model and model-S. However, no secondary flow occurs in model-L, which also indicates that the flow loss at the exit surface of the volute for model-L is obviously lower than that of the baseline model and model-S.

It is also obtained that the symmetry of streamline on the exit surface has been improved with the increase of the flow rate, and this also greatly promotes the development of secondary flow with the influence on the flow in the volute discharge.

Figure 18 shows the nine sections of volute outlet from bottom to top. Figure 19 shows the velocity distribution along the height (from 5% H to 80% H) of the volute outlet at the design condition. Plotted in Figure 19, it is observed that at the design point, the backflow area of model-L is the largest in the middle and lower part of the volute outlet. The backflow area of the model-L is reduced to a minimum at 40% H and disappears completely at 50% H, while a small range of backflow occurs in the baseline model and the model-S. The backflow completely disappears in the middle and upper part of the volute outlet for all models. At a high flow rate condition, the backflow range of all models is close to the lower part of the volute outlet. With the increase of the outlet height, the backflow range of the baseline model and model-S decreases rapidly and disappears at 50% H, while a small range of backflow still occurs in the model-L, and the velocity gradient increases in the axial direction.

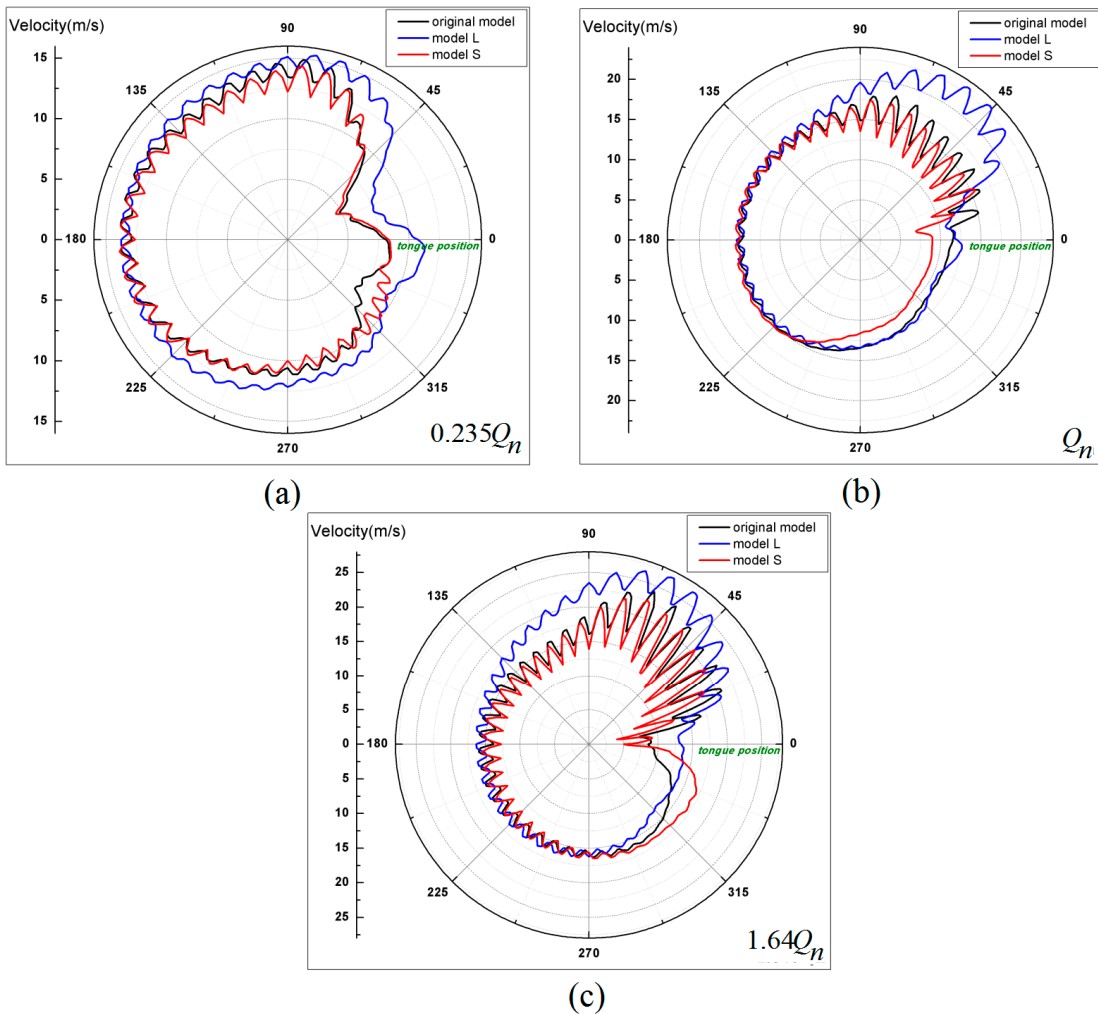

**Figure 16.** Velocity profile above the impeller. (**a**) velocity profile at 0.235 $Q_n$; (**b**) velocity profile at $Q_n$; (**c**) velocity profile at 1.64 $Q_n$.

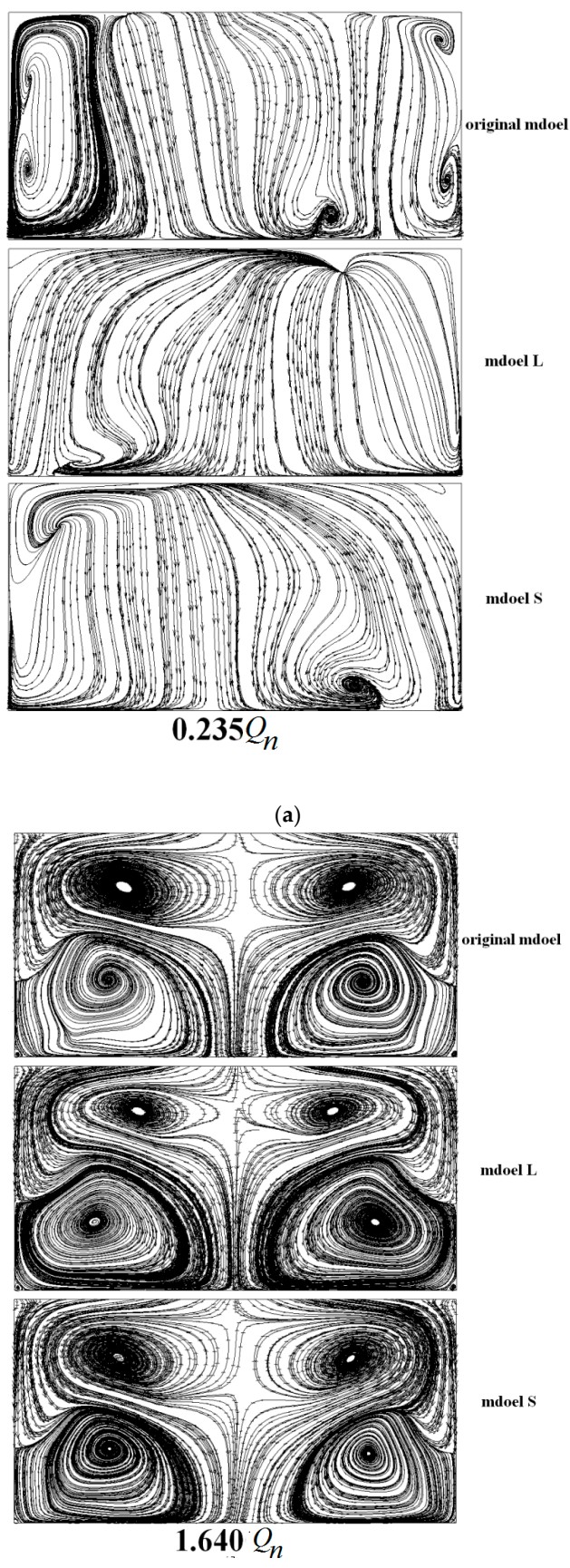

(a)

(b)

**Figure 17.** *Cont.*

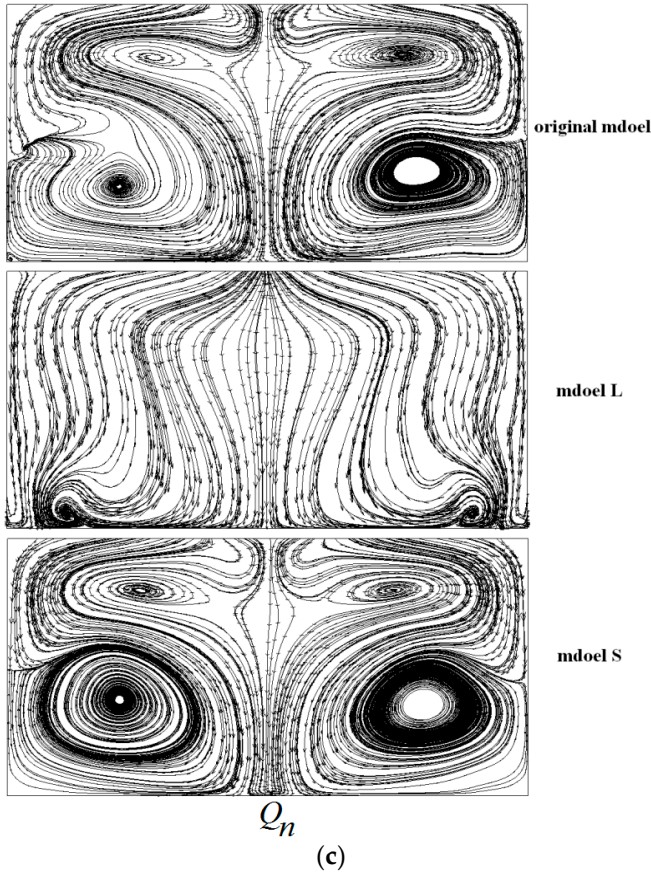

(c)

**Figure 17.** The streamline on the exit surface of volute. (**a**) at 0.235 $Q_n$; (**b**) at 1.64 $Q_n$; (**c**) at $Q_n$.

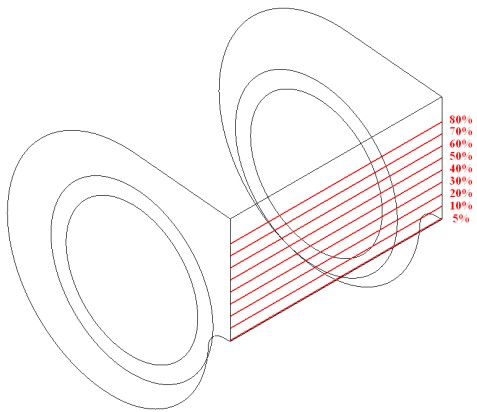

**Figure 18.** Section of volute outlet from bottom to top.

Figure 20 illustrates the velocity distribution along the height (from 5% H to 80% H) of volute outlet at the high flow rate condition ($1.64Q_n$). The x-axis refers to the axial position and the y-axis refers to the velocity perpendicular to the volute outlet. The negative value of velocity (below the 0 scale line of the ordinate as shown in Figure 19) indicates that the backflow appears at this height of the volute outlet. The backflow at the volute outlet mainly converges at the low height region of the outlet, and the range of backflow gradually narrows and finally disappears with the increase of the height.

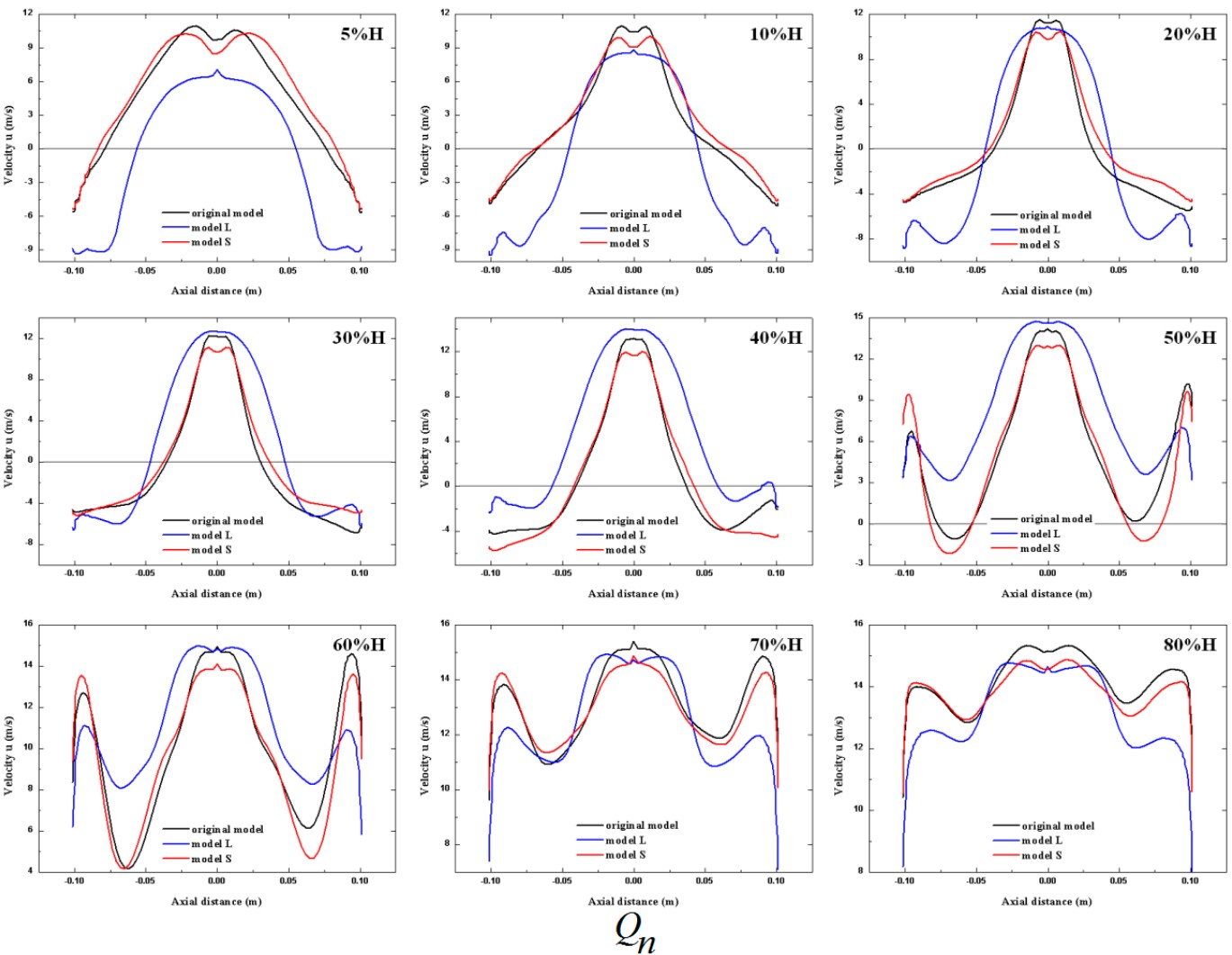

**Figure 19.** Velocity distribution along the height of volute outlet at the design condition.

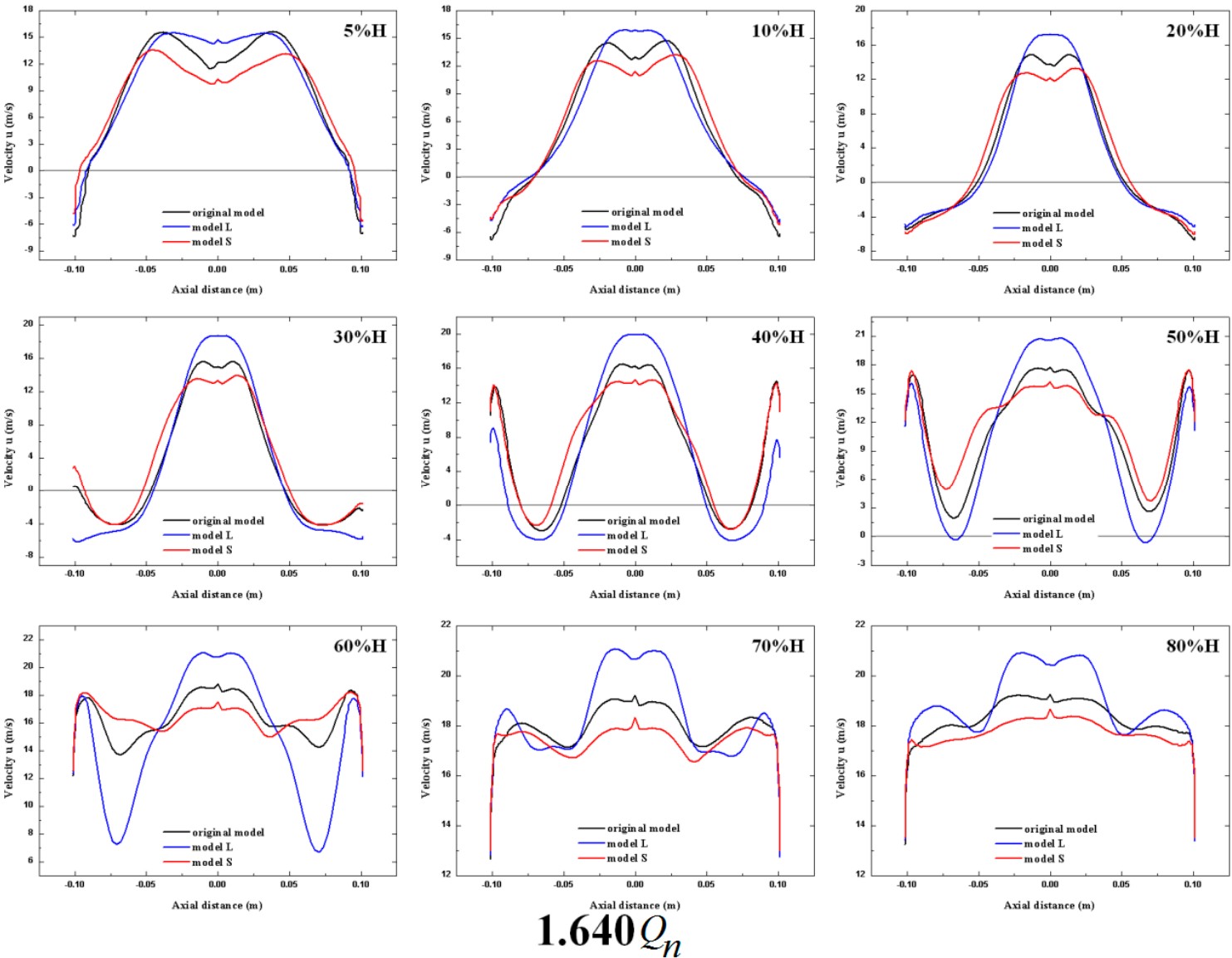

**Figure 20.** Velocity distribution along the height of volute outlet at 1.64$Q_n$.

### 4.3. Performance Results of Numerical Simulations

The relevant parameters for the performance of the fan, i.e., output power, shaft power and total-to-total efficiency for the impeller flow field, are defined by

$$SPWR = T_{imp}\omega \tag{5}$$

$$impPWR_{out} = (\Delta P_t)T_{imp}Q \tag{6}$$

$$\eta_{imp} = \frac{impPWR_{out}}{SPWR} \tag{7}$$

in which $T_{imp}$, $Q$, and $P_{imp}$ denote the impeller torque, rate of flow, total pressure increases across the bell mouth and impeller, and the fan flow rate. The performance of the entire fan is calculated differently from calculation of the impeller. The *SPWR* is again evaluated by Equation (5), but $T_{imp}$ is evaluated by integrating the torque from all the impeller blades. The lift-side total and static efficiency are defined by:

$$(\eta_t)_{lift} = \frac{(\Delta P_t)_{lift}Q_{lift}}{T_{imp}\omega}t \tag{8}$$

$$(\eta_S)_{lift} = \frac{(\Delta P_S)_{lift}Q_{lift}}{T_{imp}\omega}t \tag{9}$$

Figure 21 shows the static pressure and static pressure efficiency for various models obtained by steady numerical simulations. As shown in Figure 22, it is clearly observed that the static pressure and static pressure efficiency of model-L are obviously higher than that of model-S and the baseline model at the design flow rate, and the static pressure and static pressure efficiency of model-S are obviously lower than that of the baseline model at the design flow rate. The static pressure and static pressure efficiency obviously increases with the increase of the blade length of model-S, the baseline model and model-L at the design flow rate. It is also obtained that the static pressure value of the model-S is about 78 Pa, the static pressure value of the baseline model is about 82 Pa, and the static pressure value of the model-L is about 104 Pa. We can obtain that at the designed flow rate, the improved static pressure of the model-L rises as much as 23 Pa. It is further found that the static pressure efficiency of the baseline model is about 27.5%, and the static pressure efficiency of the model-L is about 33.5%, and also the improved static pressure efficiency of the model-L rises as much as 6% at design flow rates. Our work suggests that reducing the blade inlet angle ($\beta_{1A} > 60°$) and improving the blade outlet angle ($\beta_{2A} < 175°$) can provide a significant increase on the static pressure and the efficiency of static pressure, which improves the aerodynamic load of the forward multiblade fan to achieve the aim of energy conservation.

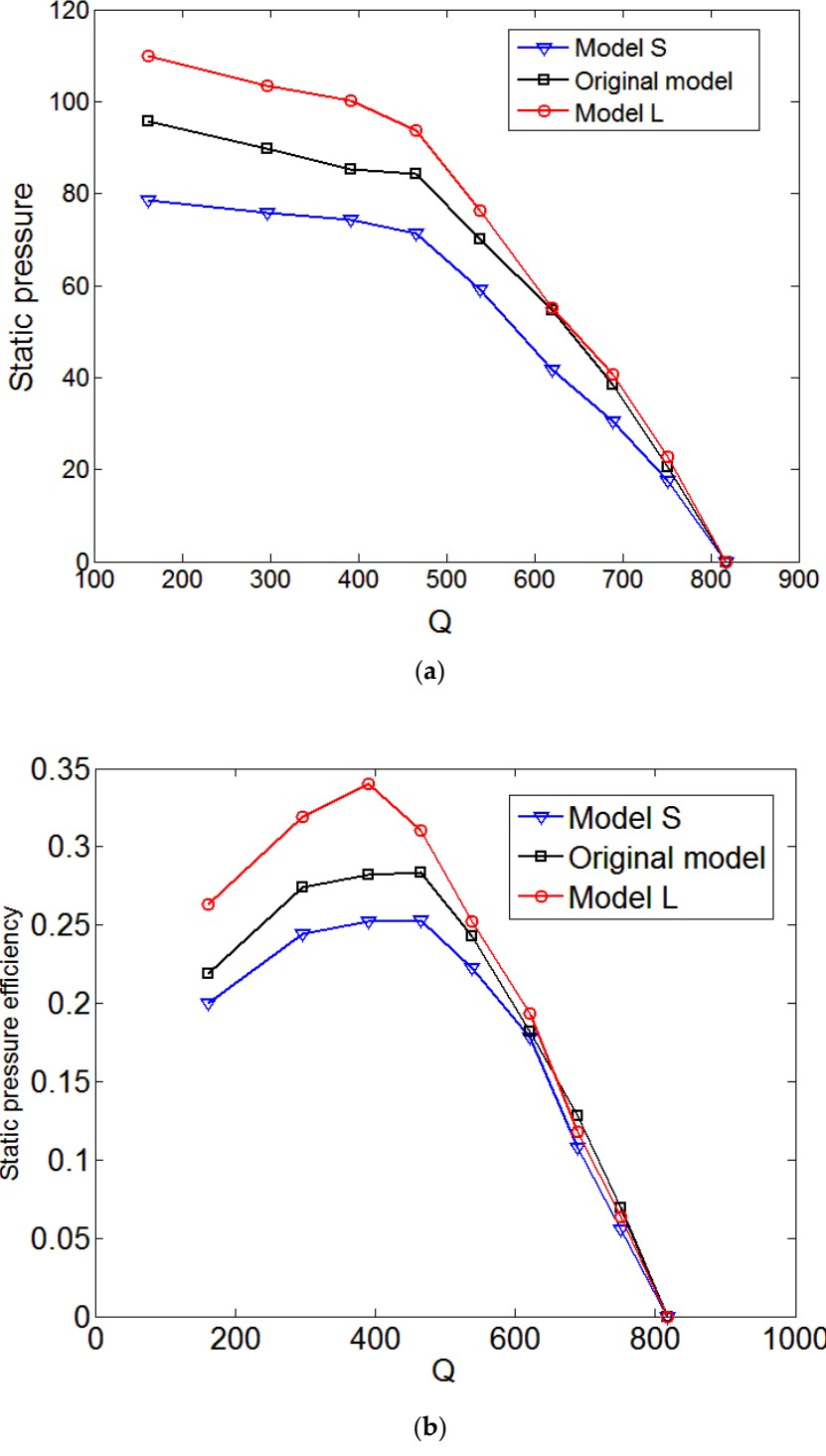

**Figure 21.** Static pressure and static pressure efficiency of different model (Q, m$^3$/h). (**a**) static pressure-flow; (**b**) static pressure efficiency-flow.

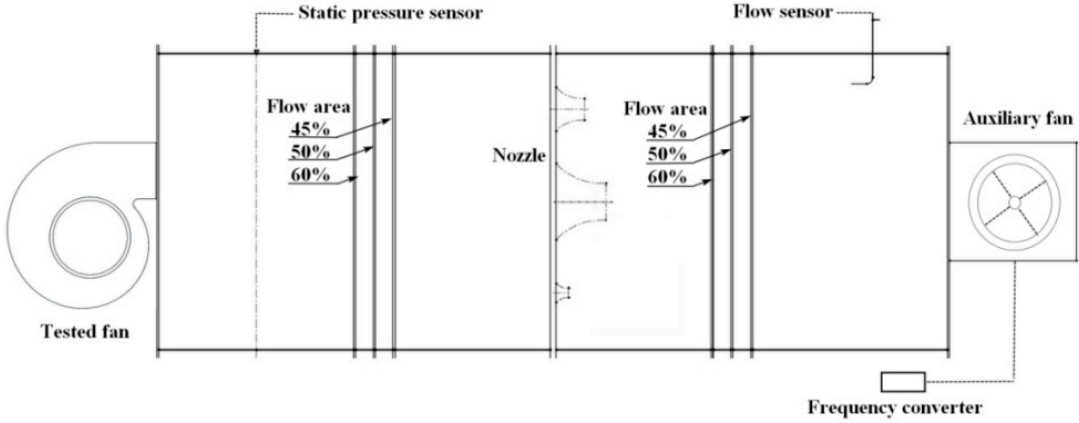

(**a**)

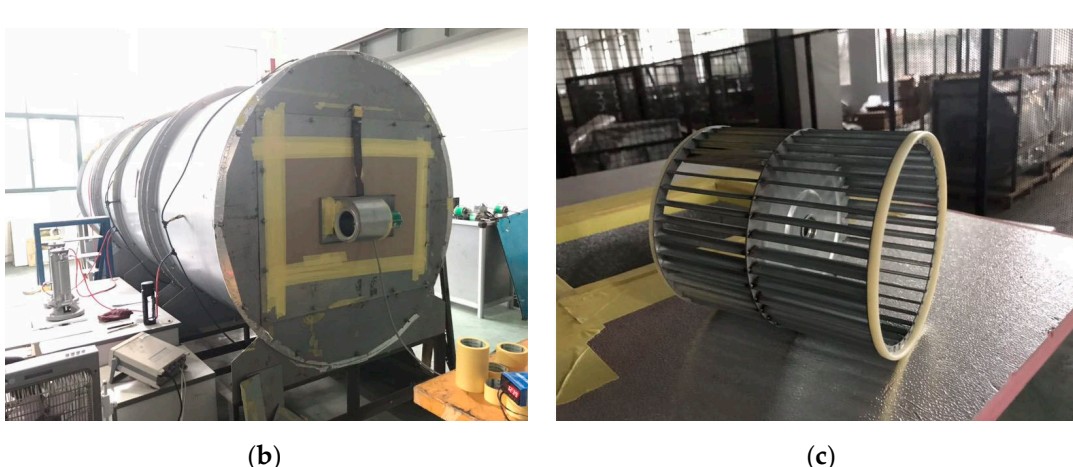

(**b**)                                              (**c**)

**Figure 22.** Experimental device and impeller model. (**a**) two-dimensional cross-section of performance measurement device; (**b**) on-site testing equipment; (**c**)impeller of model-L in experimental performance measurement.

### 4.4. Performance Results of Experimental Test

Figure 22 shows the performance measurement device of testing the centrifugal fan. Plotted in Figure 22, the centrifugal fan is mounted on the inlet of Yilida's performance measurement device. At the outlet of the wind tunnel an auxiliary fan is mounted. The network of steady flow is mounted in the chamber to measure homogeneous flow. Multiple nozzles are implemented in the chamber. The details of the test rig are shown in Figure 22b. Figure 22c presents the impeller of model-L in experimental performance measurement.

The performance characteristics of the baseline model and model-L are performed on a chamber test rig. The performance data, such as static pressure, total pressure, static pressure efficiency and total pressure efficiency, are obtained at a range of flow rates. The measuring device for testing the centrifugal fan is manufactured by Air Movement and Control Association International (AMCA) standard. The centrifugal fan is placed at the entrance of the performance measuring device, and an auxiliary fan is placed at the outlet as shown in Figure 22. A steady flow network in the chamber can provide stable flow patterns for measurement. Pairs of nozzles are installed in the chamber to obtain various flow rates. A throttling device with an auxiliary fan to control the operating point of the testing centrifugal fan is used at the outlet of the test chamber.

Figure 23 shows the static pressure and static pressure efficiency for the baseline model and model-L by experimental test. One can see that the static pressure and static pressure efficiency of

model-L are obviously higher than that of the baseline model near the design flow rate. The static pressure value of the baseline model is about 82.5 Pa, and the static pressure value of the model-L is about 105 Pa. It is obvious that compared to the baseline model, the static pressure of the model-L rises as much as 22.5 Pa at the design flow rate. It is also obtained that the static pressure efficiency of the baseline model is about 26.4% (including efficiency of the motor), and the static pressure efficiency of the model-L is about 31.5% (including efficiency of the motor), thus the static pressure efficiency of the model-L rises as much as 5% at the design flow rate.

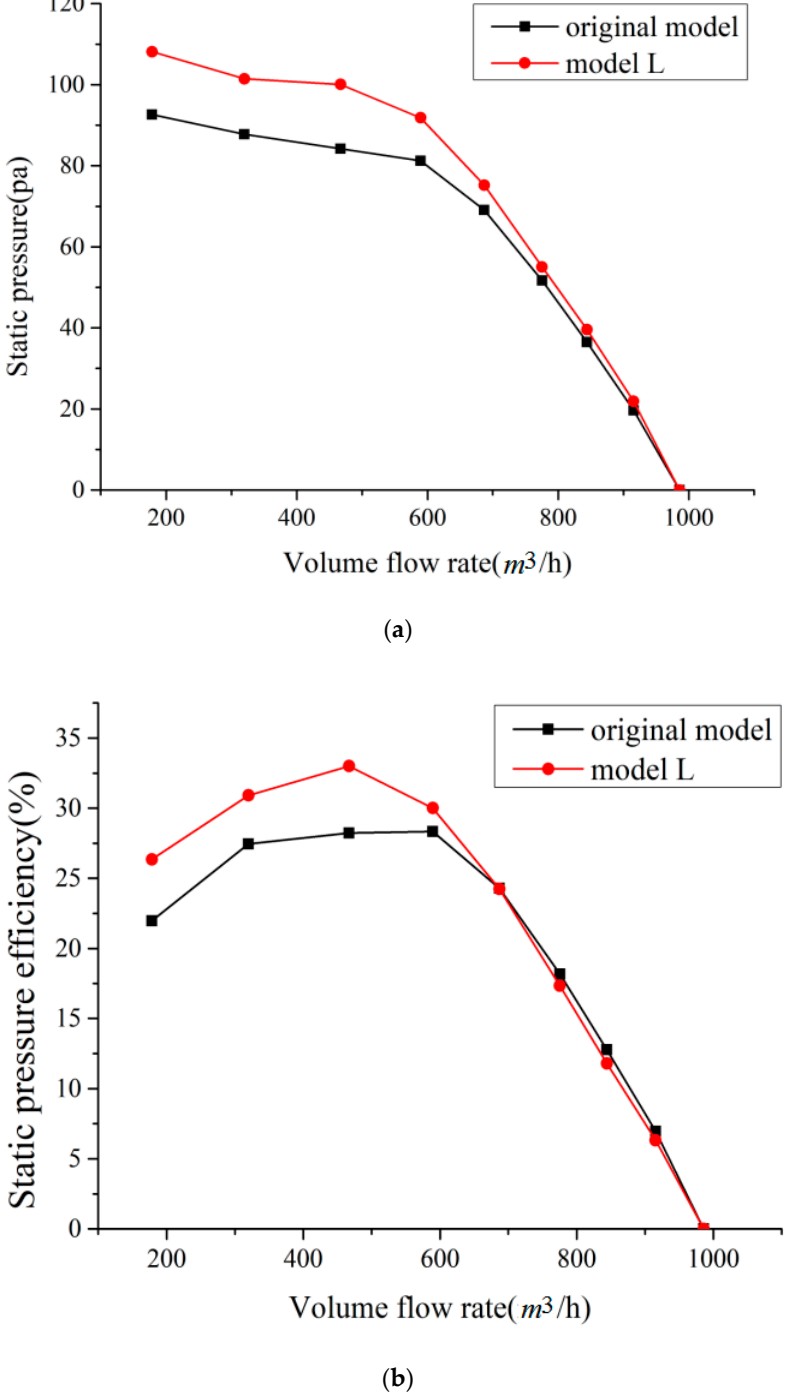

(**a**)

(**b**)

**Figure 23.** Experimental static pressure and static pressure efficiency for the baseline model and model-L. (**a**) static pressure-flow; (**b**) static pressure efficiency-flow.

Figure 24 demonstrates the total pressure and total pressure efficiency for the baseline model and model-L by experimental test. The total pressure and total pressure efficiency of model-L are obviously higher than that of the baseline model at all flow rates. The total pressure value of the baseline model is about 101.5 Pa, and the total pressure value of the model-L is about 125.4 Pa, thus the static pressure of the model-L rises as much as 25 Pa compared with that of the baseline model.

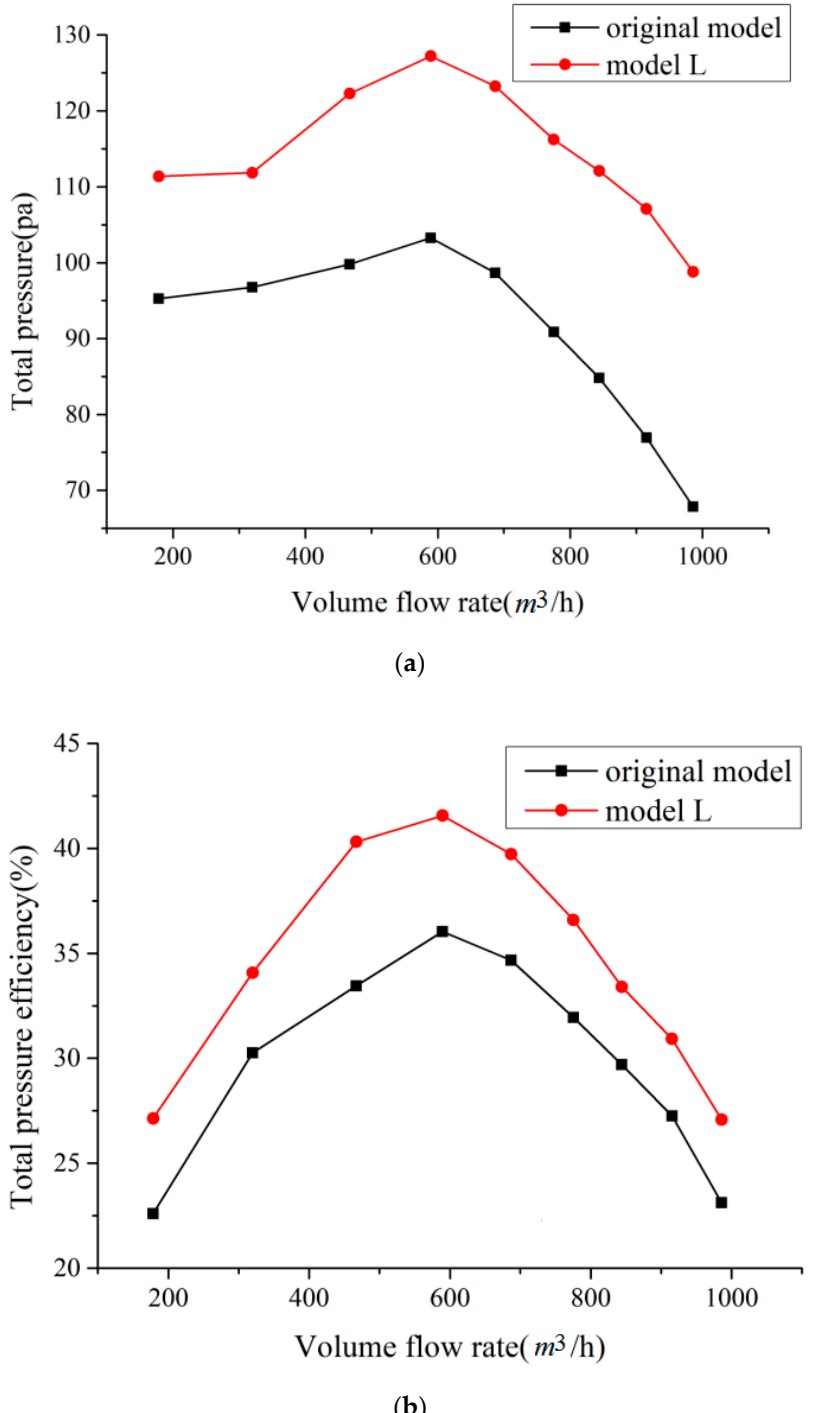

**Figure 24.** Experimental total pressure and total pressure efficiency for baseline model and model-L. (**a**) total pressure-flow; (**b**) total pressure efficiency-flow.

It is also found that the total pressure efficiency of the baseline model is about 35.5% (including the efficiency of the motor), and the total pressure efficiency of the model-L is about 41.5% (including efficiency of the motor), thus the total pressure efficiency of the model-L rises as much as 6% at the design flow rate.

## 5. Conclusions

In this paper, the effects of single-arc blade profile length on the performance of a forward multiblade fan are presented and demonstrated. The internal complex flow characteristics of fans are investigated by numerical simulations by implementing the RANS turbulence model, and the performance of fans is studied by numerical simulations and experimental test. Several conclusions can be summarized as follows:

The gradient of the absolute velocity among the blades in model-L is clearly lower than that of the baseline model and model-S, and the gradient of the absolute velocity among the blades decreases in model-L, which brings about the size of the flow separation among the blades.

At the design point $Q_n$, the area of turbulent kinetic energy at the inlet of the volute for model-L is smaller than that of the baseline model and model-S. The flow loss at the exit surface of the volute for model-L is obviously lower than that of the baseline model and model-S

Experimental results demonstrate that the static pressure of model-L rises as much as 22.5 Pa and 26.2%, while the static pressure efficiency of the model-L rises as much as 5% at the design flow rate. Meanwhile, the total pressure of the model-L rises as much as 25 Pa and 23.6%, and the total pressure efficiency of the model-L rises as much as 6% at the design flow rates.

It is found that a properly increasing blade working area provides a significant increasing on the static pressure, total pressure, the efficiency of static pressure and total pressure efficiency. The increase of the blade working area obviously improves the aerodynamic load of fans to achieve the purpose of energy conservation.

**Author Contributions:** The following statements could be used Y.W. and C.Y. conceived and designed the experiments; J.X. performed the experiments; W.C. and Z.W. analyzed the data; Z.Z. contributed reagents/materials/analysis tools; Y.W. wrote the paper.

**Funding:** This work was supported by the National Natural Science Foundation of China (11872337, 11902291 and U1709209), Natural Science Foundation of Zhejiang Province (LY18A020010), public welfare technology application research project of Zhejiang province (2017C31075), and Fundamental Research Funds of Zhejiang Sci-Tech University (2019Y004).

**Acknowledgments:** The authors appreciate sincerely the referees' valuable comments and suggestions on our work.

**Conflicts of Interest:** The authors declare no conflict of interest.

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
