# Peer review of "Effects of Single-arc Blade Profile Length on the Performance of a Forward Multiblade Fan"

_processes, doi:10.3390/pr7090629_

Round 1

Reviewer 1 Report

General Comments

The paper needs to be thoroughly edited by a native English speaker. I began correcting up to Line 29 (see below) but then stopped because there were too many mistakes and would take up too much time to make note of each error. Much more information is required on the experimental and numerical set-up and methods. It is concerning to see that the blade inlet angles are quoted as accurate to +/- 100th of a degree, lengths of the blade profiles as accurate to +/- 1 micrometer and the flow rates accurate to +/- 0.1 grams/second. This would indicate that very little thought or thoroughness has been applied to the analysis and tracking of errors (such as measurement errors) in this work.

Specific Comments

Abstract

Referring to “model L” and “model S” has no meaning to someone reading the abstract. Please rewrite, so that the abstract is self-sufficient and does not require a reader to download the entire paper to see what you are referring to. Lines 11-14 : The following sentence does not make sense, please re-write: “The present work emphasizes that the use of properly reducing blade inlet angle(β1A) and properly improving blade outlet angle (β2A) is to increase blade profile length, suggests that a good physical understanding of internal complex flow characteristics and the aerodynamic performance of fan” Line 18: “shows” = show

1. Introduction

Line 28: “has” = have Line 28: ‘widely adopted industrial application” = widely adopted in industrial applications Line 29: “...systems, home appliance...” = ..systems and home appliance...

2. Governing Equations and Numerical Method

Line 65: You do not need to include a subsection, when there is only one. It only makes sense to split a section into different subsections when there are different subsections (i.e. more than 1). Lines 69-71: You have used a different font for \rho in the Equations 1 and 2, and in the text in line 71. Please be consistent. (The same is true for the other variables x, u, P and f). If using LaTeX, simply surround the variable with $ to enter into math mode. i.e. “in which $\rho$ is density, $x_i$ and $x_j$ denote....” Lines 76-80: The same as for Equations 1 and 2 above. Be consistent with the fonts, also, you have not set ‘M’ as subscript in ‘YM’ Equations 5 -6: These variables and parameters are hard to read. Is “SPWR” mean to be shaft power? And “impPWR_{out}” meant to be output impeller power? I would suggest using ‘P’ for power and then a subscript to denote where the power is measured e.g. P_s is shaft power and P_{out} is the output power. This section only detailed the governing equations and gave no details on the numerical method of how they are solved. Please include an additional subsection with “Numerical methods” (i.e. 2.1 Governing Equations of Fluid Flow. 2.2 Numerical methods)

3. Experiment Scheme and Experiment Method

Line 65: You do not need to include a subsection, when there is only one. It only makes sense to split a section into different subsections when there are different subsections (i.e. more than 1). Line 98: You have claimed the length of the arc is 10.904mm? Is the length of the arc really accurate to +/- 1 micrometer? Line 113-123: This information and Figures 3 and 4, belong in Section 2, when discussing the numerical method. Much more information is needed on the experimental set-up and procedures. What was measured, where was it measured and how was it measured, for example, would be a good start.

4. Results and Discussions

Lines 125 – 130: This is not Results, this information belongs in Section 2, when discussing the numerical method.# Lines 143-144: Was the mass flow rate really accurate to +/- 0.1 grams per second? Line 159: There is a strange one sentence paragraph here. This should be merged with the following paragraph. Line 278: This should be Section 4.3, you have already defined Section 4.2 on Line 141 Line 297: This should be Section 4.4 Lines 298 – 312: This is not Results and should be included in Section 3 when describing the Experiment set up Figures 23 and 24: The meters cubed in the x-label should be superscript i.e. $m^3$ Why wasn’t Model S tested experimentally?

Author Response

Modification Of No. Processes-19-573209

Reviewer1:

The paper needs to be thoroughly edited by a native English speaker. I began correcting up to Line 29 (see below) but then stopped because there were too many mistakes and would take up too much time to make note of each error. Much more information is required on the experimental and numerical set-up and methods. It is concerning to see that the blade inlet angles are quoted as accurate to +/- 100th of a degree, lengths of the blade profiles as accurate to +/- 1 micrometer and the flow rates accurate to +/- 0.1 grams/second. This would indicate that very little thought or thoroughness has been applied to the analysis and tracking of errors (such as measurement errors) in this work.

Response:  Corrected.

 Firstly, the authors appreciate sincerely the referees valuable comments and suggestions on this work.  English grammar and errors have been completely corrected and English writing has been improved greatly in the revised manuscript.

Specific Comments

Abstract

Referring to “model L” and “model S” has no meaning to someone reading the abstract. Please rewrite, so that the abstract is self-sufficient and does not require a reader to download the entire paper to see what you are referring to. Lines 11-14 : The following sentence does not make sense, please re-write: “The present work emphasizes that the use of properly reducing blade inlet angle(β1A) and properly improving blade outlet angle (β2A) is to increase blade profile length, suggests that a good physical understanding of internal complex flow characteristics and the aerodynamic performance of fan” Line 18: “shows” = show

Response:  Corrected.

The present work emphasizes that the use of properly reduced blade inlet angle(β1A) and properly improved blade outlet angle (β2A) is to increase blade profile length, which suggests a good physical understanding of internal complex flow characteristics and the aerodynamic performance of fan. Numerical results indicate that the gradient of the absolute velocity among the blades in model-L (reducing the blade inlet angle and improving blade outlet angle) is clearly lower than that of the baseline model and model-S (improving the blade inlet angle and reducing blade outlet angle), a number of secondary flows arise on the exit surface of baseline model and model-S. However, no secondary flow occurs in model-L, and the flow loss at the exit surface of volute for model-L is obviously lower than that of the baseline model at design point. 

Introduction

Line 28: “has” = have Line 28: ‘widely adopted industrial application” = widely adopted in industrial applications Line 29: “...systems, home appliance...” = ..systems and home appliance...

Response:  Corrected.

Governing Equations and Numerical Method

Line 65: You do not need to include a subsection, when there is only one. It only makes sense to split a section into different subsections when there are different subsections (i.e. more than 1). Lines 69-71: You have used a different font for \rho in the Equations 1 and 2, and in the text in line 71. Please be consistent. (The same is true for the other variables x, u, P and f). If using LaTeX, simply surround the variable with $ to enter into math mode. i.e. “in which $\rho$ is density, $x_i$ and $x_j$ denote....” Lines 76-80: The same as for Equations 1 and 2 above. Be consistent with the fonts, also, you have not set ‘M’ as subscript in ‘YM’ Equations 5 -6: These variables and parameters are hard to read. Is “SPWR” mean to be shaft power? And “impPWR_{out}” meant to be output impeller power? I would suggest using ‘P’ for power and then a subscript to denote where the power is measured e.g. P_s is shaft power and P_{out} is the output power. This section only detailed the governing equations and gave no details on the numerical method of how they are solved. Please include an additional subsection with “Numerical methods” (i.e. 2.1 Governing Equations of Fluid Flow. 2.2 Numerical methods)

Response:  Corrected.

Experiment Scheme and Experiment Method

Line 65: You do not need to include a subsection, when there is only one. It only makes sense to split a section into different subsections when there are different subsections (i.e. more than 1). Line 98: You have claimed the length of the arc is 10.904mm? Is the length of the arc really accurate to +/- 1 micrometer? Line 113-123: This information and Figures 3 and 4, belong in Section 2, when discussing the numerical method. Much more information is needed on the experimental set-up and procedures. What was measured, where was it measured and how was it measured, for example, would be a good start.

Response:  Corrected.

The blade profile of baseline model is a single circular arc with the length of 10.904 mm, model-L refers to the model with a longer blade and model-S with a shorter blade. Plotted in Figure 2, it is seen that the blade profile of model-L (reducing the blade inlet angle and improving blade outlet angle) is 150% that of the baseline model and model-S(improving the blade inlet angle and reducing blade outlet angle) is 80%. The blade profiles of both modified models are still single circular arc

The RNG k- epsilon model adds a condition to the torsional equation, which takes into account the turbulent vortex and provides an analytical formula for the viscosity of flows with low Reynolds number, which can obtain flow analysis data with higher reliability and accuracy. The previous studies on similar simulation are implemented in fluid flow analysis of the computational domain[15,16]. ANSYS CFX is implemented in this paper.

[15]Montazerin N,, Damangir A, and Mirzaie H, Inlet induced flow in  squirrel-cage fans. Journal of Power and Energy,  2000,214:243-253,.

[16]Bayomi N N,  and Osman A M, Effect of inlet straighteners on centrifugal fan performance. Energy Conversion and Management, 2006,47:3307-3318,.

The centrifugal fan is mounted on the inlet of Yilida’s performance measurement device. At the outlet of wind tunnel an auxiliary fan is mounted. The network of steady flow is mounted in the chamber to measure homogeneous flow. Multiple nozzles are implemented in the chamber. The details of the test rig are shown in Figure 22(b). Figure 22 (c) presents the impeller of model-Lin experimental performance measurement. The performance characteristics of the baseline model and model-L are performed on a chamber test rig.

Results and Discussions

Lines 125 – 130: This is not Results, this information belongs in Section 2, when discussing the numerical method.# Lines 143-144: Was the mass flow rate really accurate to +/- 0.1 grams per second? Line 159: There is a strange one sentence paragraph here. This should be merged with the following paragraph. Line 278: This should be Section 4.3, you have already defined Section 4.2 on Line 141 Line 297: This should be Section 4.4 Lines 298 – 312: This is not Results and should be included in Section 3 when describing the Experiment set up Figures 23 and 24: The meters cubed in the x-label should be superscript i.e. $m^3$ Why wasn’t Model S tested experimentally?

Response:  Corrected.

The above equations are solved by implementing finite volume method. In first, the time discretization of the Navier–Stokes equations coupled with some governing equation for a pollutant is implemented in this paper. The spatial schemes are used to account for convective terms, viscous terms, mean pressure gradient effects and source terms associated with momentum or scalar equations, focusing on triangular meshes [25,26]. Finite element methods[21,22,23,24] and finite volume method[25,26,] are very effective tool to solve some partial differential equations (PDEs) on complex geometries , which is applied in a wide range of engineering and biomedical disciplines[27,28,29,30].  

 The centrifugal fan is mounted on the inlet of Yilida’s performance measurement device. Meanwhile, an auxiliary fan is installed on the outlet of measurement device. The network of steady flow is installed in the chamber to measure homogeneous flow. Multiple nozzles are adopted in the chamber. The monicker and position of each instrument are represented in Figure6, respectively. The details of the test rig are shown in Figure 22. Figure 22 shows the drum performance test device and various instruments used in the testing process of a centrifugal fan, such as the speed adjustment instrument, pressure collector and so on.

Reviewer 2 Report

Indices of references as well as figures are not correct The structure of the document shall be corrected, escepially the description of the numerical model as well as the post processing is very unclear; additionally the description of the experimental setup is rather incomplete Information on the numerical meshes is insufficient, a mesh independency study is missing; The description of the numerical setup should be improved; e.g. which MFR-interface model was applied Information on the measurement uncertainty of the experimental setup is missing Proof reading is strongly recommended

Author Response

Modification Of No. Processes-19-573209

Reviewer2:

Comments and Suggestions for Authors

Indices of references as well as figures are not correct. The structure of the document shall be corrected, the description of the numerical model as well as the post processing is very unclear;

Response:  Corrected.

additionally the description of the experimental setup is rather incomplete Information on the numerical meshes is insufficient, a mesh independency study is missing;

Response:  Corrected.

As shown in Figure 4(b), we can clearly see that the change rate of static pressure is less than 1% at different mesh numbers. Thus, we can ignore the static pressure change. The calculation results are not effected by the number of grids.

(b)

Figure 4 Static pressure at different number of grids

The description of the numerical setup should be improved; e.g. which MFR-interface model was applied Information on the measurement uncertainty of the experimental setup is missing Proof reading is strongly recommended.

Response:  Corrected.

CFD results were implemented to provide insight into revealing flow loss in internal flow of fan. Thus, the prediction method of CFD is represented following the internal flow mechanisms in fan. The section of computational methodology includes solver descriptions of the Navier-Stokes and the  used turbulence models associated, computational procedures for nonrotating and rotating components of the fan, gridding strategy and a study of grid density, and flow parameters for performance of predicting fan. To validate the predictions CFD of for the performance of fan data, comparisons with 1/5-scale model fan test dataare provided.

The above equations are solved by implementing finite volume method. In first, the time discretization of the Navier–Stokes equations coupled with some governing equation for a pollutant is implemented in this paper. The spatial schemes are used to account for convective terms, viscous terms, mean pressure gradient effects and source terms associated with momentum or scalar equations, focusing on triangular meshes [25,26]. Finite element methods[21,22,23,24] and finite volume method[25,26,] are very effective tool to solve some partial differential equations (PDEs) on complex geometries , which is applied in a wide range of engineering and biomedical disciplines[27,28,29,30].  

The centrifugal fan is mounted on the inlet of Yilida’s performance measurement device. At the outlet of wind tunnel an auxiliary fan is mounted. The network of steady flow is mounted in the chamber to measure homogeneous flow. Multiple nozzles are implemented in the chamber. The details of the test rig are shown in Figure 22(b). Figure 22 (c) presents the impeller of model-L in experimental performance measurement. The performance characteristics of the baseline model and model-L are performed on a chamber test rig. 

Reviewer 3 Report

Detailed comments are listed below.

Why did the author use CFD? please describe the reason in the background. Also, add more references related to CFD investigation. Please make an annotation on Fig. 1. Where is hub plate, volute, etc What did the author mean to the original model? Please state clearly. For example, the original model is referred to the previous work by ..... []   Please make an annotation on Fig. 2. Where is b, B, etc Please describe the blade inlet angle and outlet angle by picture. How to get and define the optimum of the blade angle in model S and L?  Why did the author choose RNG k-epsilon model? Did the author compare with other turbulence models? It is better also the refer the previous study on similar simulation. What CFD software did the author use? Figure 6 does not show the result on the absolute velocity. Please revise line 149, p.6 Please revise line 159, p.6. Legend is too small. Please make it bigger.

Author Response

Modification Of No. Processes-19-573209

Reviewer3:

Detailed comments are listed below.

Why did the author use CFD? please describe the reason in the background. Also, add more references related to CFD investigation. Please make an annotation on Fig. 1. Where is hub plate, volute, etc What did the author mean to the original model? Please state clearly. For example, the original model is referred to the previous work by ..... []   Please make an annotation on Fig.

Response:  Corrected.

CFD results were implemented to provide insight into revealing flow loss in internal flow of fan. Thus, the prediction method of CFD is represented following the internal flow mechanisms in fan. The section of computational methodology includes solver descriptions of the Navier-Stokes and the  used turbulence models associated, computational procedures for nonrotating and rotating components of the fan, gridding strategy and a  study of grid density, and flow parameters for performance of predicting fan. To validate the  predictions CFD of for the performance of fan data, comparisons with 1/5-scale model fan test dataare provided. The complex vortex structure of the baseline model on the internal flow and performance is referred to the previous work by Lun [13].

[13] Lun Y X, Lin LM, Zhu ZC,Wei Y K,Effects of Vortex Structure on Performance Characteristics of a Multiblade Fan with Inclined tongue,Proceedings of the Institution of Mechanical Engineers Part A Journal of Power and Energy, 2019,DOI: 10.1177/0957650919840964

[18]Hosangadi, A, Lee, R A, York, B J, Sinha, N, and Dash, S M, ,Upwind Unstructured Scheme for Three-Dimensional Combusting Flows, J.Propul. Power, 1996,12 : 494–502.

[19] Hosangadi, A , Lee, R A, Cavallo, P A, Sinha, N. and York, B J. Hybrid, Viscous, Unstructured Mesh Solver for Propulsive Applications, AIAA 34th JPC, Cleveland, OH, Paper. AIAA,1998,98:3153-3152.

[20]Lee, Y T, Mulvihill, L, Coleman, R, Ahuja, V, Hosangadi, A, Birkbeck, R,Becnel, A, and Slipper, M, LCAC Lift Fan Redesign and CFD Evaluation, Naval Surface Warfare Center Report,2007 NSWCCD-50-TR-2007/031.

Where is b, B, etc Please describe the blade inlet angle and outlet angle by picture. How to get and define the optimum of the blade angle in model S and L?  

Response:  Corrected.

The blade profile of baseline model is a single circular arc with the length of 10.904 mm, model-L refers to the model with a longer blade and model-S with a shorter blade. Plotted in Figure 2, it is seen that the blade profile of model-L (reducing the blade inlet angle and improving blade outlet angle) is 150% that of the baseline model and model-S(improving the blade inlet angle and reducing blade outlet angle) is 80%. The blade profiles of both modified models are still single circular arc

Why did the author choose RNG k-epsilon model? Did the author compare with other turbulence models? It is better also the refer the previous study on similar simulation. What CFD software did the author use?

Response:  Corrected.

The RNG k- epsilon model adds a condition to the torsional equation, which takes into account the turbulent vortex and provides an analytical formula for the viscosity of flows with low Reynolds number, which can obtain flow analysis data with higher reliability and accuracy. The previous studies on similar simulation are implemented in fluid flow analysis of the computational domain[15,16]. ANSYS CFX is implemented in this paper.

[15]Montazerin N,, Damangir A, and Mirzaie H, Inlet induced flow in  squirrel-cage fans. Journal of Power and Energy,  2000,214:243-253,.

[16]Bayomi N N,  and Osman A M, Effect of inlet straighteners on centrifugal fan performance. Energy Conversion and Management, 2006,47:3307-3318,.

Figure 6 does not show the result on the absolute velocity. Please revise line 149, p.6 Please revise line 159, p.6. Legend is too small. Please make it bigger.

Response:  Corrected.

Round 2

Reviewer 1 Report

I do not believe the comments from the review have been adequately addressed.

Author Response

We  sincerely appreciate  your valuable comments and suggestions on this work. We have done our best to revise our paper.  The bright innovation of this paper is that the increasing blade working area provides significant physical insight into increasing the static pressure, total pressure, the efficiency of static pressure, and total pressure efficiency. Our work indicates that the use of properly reduced blade inlet angle(β1A) and properly improved blade outlet angle (β2A) is to increase the length blade profile,which suggests a good physical understanding of internal complex flow characteristics and the aerodynamic performance of fan. Numerical results indicate that the gradient of the absolute velocity among the blades in model-L (reducing the blade inlet angle and improving blade outlet angle) is clearly lower than that of the baseline model and model-S (improving the blade inlet angle and reducing blade outlet angle), a number of secondary flows arise on the exit surface of baseline model and model-S. However, no secondary flow occurs in model-L, and the flow loss at the exit surface of volute for model-L is obviously lower than that of the baseline model at design point. The comparison of the test results further shows that to improve the blade profile length is to increase the static pressure and the efficiency of static pressure, improved static pressure of the model-L rises as much as 22.5 Pa and 26.2%, improved static pressure efficiency of the model-L rises as much as 5 % at design flow rates. 

Reviewer 2 Report

The structure of the document still need some improvement;

e.g:
rearrangement of equations 5 to 9 into results section; there are two sections 4.2; Figure 22 in line 292 should be read as Figure 21?

Author Response

We appreciate sincerely the refereesvaluable comments and suggestions on this work. 

1. rearrangement of equations 5 to 9 into results section;

 Response:  Corrected.

2. there are two sections 4.2;

 Response:  Corrected.

3. Figure 22 in line 292 should be read as Figure 21?

 Response:  Corrected.

Reviewer 3 Report

The authors have shown their effort to revise the manuscript.

Author Response

Thanks a lot.  the revised manuscript has been made great efforts.